# Development of Algorithms for Identifying Parameters of the Maritime Vessel Motion Model in Operating Conditions with Elements of Intellectual Analysis

**Nikolay Ivanovskii [1], Sergei G. Chernyi [1,2,3,*], Anton Zhilenkov [2] and Vitalii Emelianov [4]**

1.  Department of Navigation and Industrial Fishing, Kerch State Maritime Technological University, 298309 Kerch, Russia; inv8@mail.ru
2.  Department of Cyber-Physical Systems, St. Petersburg State Marine Technical University, 190121 Saint-Petersburg, Russia; zhilenkovanton@gmail.com
3.  Department of Complex Information Security, Admiral Makarov State University of Maritime and Inland Shipping, 198035 Saint-Petersburg, Russia
4.  Business Informatics Department, Financial University under the Government of the Russian Federation, 49 Leningradsky Prospekt, 125993 Moscow, Russia; v.yemelyanov@gmail.com
*   Correspondence: sergiiblack@gmail.com

**Abstract:** The article examines the synthesis of algorithms for the estimation of the random parameters of ship movement models, based on measured information in field tests. In addition, accuracy analysis of the synthesized algorithms is provided. The derived algorithms are relatively simple and allow highly precise unknown parameters for estimation of ship motion models at the non-real-time scale to be obtained using the measurements recorded in field tests. The results can be used in the construction of automated ship control systems, or in the development of navigation simulators and the creation of ship models.

**Keywords:** safety; navigation; risk assessment; vessel; mathematical model; control; identification; maritime; random parameter





## 1. Introduction

Algorithm elaboration for the automated control system of a specific vessel is carried out using the vessel's movement model. The ship motion model parameters can significantly affect the ship control algorithm [1]. When using the ship motion model, the question about the correspondence of the modeled phase coordinates of the vessel motion to the real ship motion arises. A number of ship motion model parameters are set in the form of a priori values, hence clarification based on field information is required when testing the automated control system. To solve this problem, it is necessary to synthesize algorithms for the estimation of motion model a priori parameters from the full-scale information of the ship's motion. The problem of estimating entrained water and the ship's inertia moment was considered in [2,3].

## 2. Research Objective

The mathematical model of the ship's motion as a controlled dynamic system can generally be represented as follows:

$$S(t) = F(t, C, S(0), U(t), L(t), E(t))$$

where F—is the operator that characterizes this particular mathematical model;

C—is the vector of constant system parameters that characterize this particular simulated vessel;

S(t)—is the set of variable parameters describing the state of the system at a moment in time $t$.

If we consider the plane-parallel movement of the ship we can restrict ourselves to three parameters—coordinates $x_0$ and $y_0$, and heading angle q:

$$S(t) = (x_0(t), y_0(t), q(t))$$

where U($t$) represents the control actions on the system at different times: rudder angle $\delta_R(t)$, rotation frequency $n_m(t)$, propeller step ratio H/D($t$), and thruster control position $N_{thruster}$ ($t$), which sets its relative power as a percentage of the maximum possible:

U($t$) = ($\delta_R(t)$, $n_m(t)$, H/D($t$), $N_{thruster}$ ($t$)).

L($t$)—is the system load function, in this case the distribution of all cargoes of the ship;

E($t$)—is the function of external disturbing influences on the system: depths at all points of the water area; wind and current speed and direction; amplitude and phase spectra of waves; and the spectra of directions of wave propagation at all frequencies for all points of the water area at all times.

The following classes of problems are well-known from theory and are associated with this mathematical model:

(1) Direct modeling problem: Required to determine the evolution of the modeled system, the law of variation of variable parameters S($t$) in time with known F, C, U($t$), L($t$), E($t$). The direct modelling problem is a formalization of the "what if?" question.

(2) Inverse problems: The requirement is to determine what was or should be inputs to the system, so that the output is a specific behavior of the system S($t$). The inverse problem is a formalization of the "how to do it?" question.

The following cases of inverse problems are notable:

(2.1) C = ? with known F, S($t$), U($t$), L($t$), E($t$). This is the task of designing a new controlled dynamic system—in this case, a new vessel with the required qualities.

(2.2) F = ? with known C, S($t$), U($t$), L($t$), E($t$). This is the task of constructing a new empirical mathematical model based on an already existing real controlled dynamic system.

(2.3) U($t$) = ? with known C, F, S($t$), L($t$), E($t$). This is the task of constructing an adaptive control algorithm (control system, control device). In addition, this class includes the task of predicting the movement of the vessel for the feasibility of the required maneuver.

(2.4) L($t$) = ? with known C, F, S($t$), U($t$), E($t$) or E($t$) = ? with known C, F, S($t$), U($t$), L($t$). This is the problem of indirectly measuring the load on the system or external disturbing influences through the identification of changes (disturbances) in the behavior of the controlled system.

In the framework of the current study, the direct problem of modeling and identification of the parameters of the ship model is mainly investigated.

As a basic model, this study used a mathematical model of the ship's movement. Based on the materials from Lloyd's Register (https://www.lr.org/en/ accessed on 1 February 2021) and International Maritime Organization (IMO) (https://www.imo.org/ accessed on 5 January 2021), conclusions can be drawn about the relevance of mathematical models. Well-known contributions to the research have been made by authors including Voytkunsky et al., 1973; Hoffman, 1988; Pavlenko, 1979; Sobolev, 1976; Tumashik, 1978; Fedyaevsky and Sobolev, 1963. Leading maritime organizations have recognized these mathematical models and the early works that are used in basic global research. On the basis of these fundamental models, mathematical structures have been tested that validate the accuracy and applicability of the models. Based on the information provided, the model can be justified as follows:

$$\frac{dv_x}{dt} = \frac{\omega(m_{22}v_y + l_{26}\omega)}{m_{11}} - \frac{C_r(\delta-\beta)^2(v^2+\omega^2 L_r^2)\rho}{2m_{11}} + \left(\frac{n_v^2}{m_{11}}\right)T_x + \left(\frac{n_v}{m_{11}}\right)vK_v + \frac{C_x(C_{x0},\beta)\rho S_{cp}v_2^2}{2m_{11}} \tag{1}$$

$$\frac{dv_y}{dt} = -\frac{\omega m_{11} v_x}{m_{22}} - \left( \begin{array}{c} C_\beta \beta \left(1 - \overline{\omega}^2\right) + C_{\omega\beta} \overline{\omega}\beta - \\ -C_{\omega\omega} \overline{\omega} \left|\overline{\omega}\right| \frac{A_p \left(v^2 + \omega^2 L_r^2\right)\rho}{2m_{22}} \\ -m_y S_r (\delta - \beta)\left(v^2 + \omega^2 L_r^2\right) \end{array} \right) \rho/2m_{22}$$

$$\frac{d\omega}{dt} = \frac{v_x (m_{11} - m_{22})\left(v_y + \frac{\omega l_{26}}{m_{22}}\right)}{J_Z} +$$

$$+ \frac{\left(C_{m\beta}\beta v^2 - C_{m\omega}\overline{\omega}\left(v^2 + \omega^2 L_r^2\right)\right) A_p L \rho}{2J_Z} +$$

$$+ \frac{m_y S_r (\delta - \beta)\left(v^2 + \omega^2 L_r^2\right) L_r \rho}{2J_Z}$$

where $v(t) = \left(v_x^2 + v_y^2\right)^{\frac{1}{2}}$, $\overline{\omega} = \frac{\omega L}{\left(v^2 + \omega^2 L^2\right)^{\frac{1}{2}}}$, $\beta(t) = -\arcsin\left(\frac{v_y(t)}{v(t)}\right)$, $C_x(C_{x0}, \beta) = -0,075$ $\sin((\pi - \arcsin(C_{x0}/0,075))(1 - |\beta|))$, $C_{m\beta}(\beta) = m_1 \sin(2\beta) + m_2 \sin(\beta)$.

The standard notation for the $x$ and $y$ axes is used for 2D space. The estimation shown in Figure 1 is defined as $x = Vx$ [m/s], $y = t$ [s].

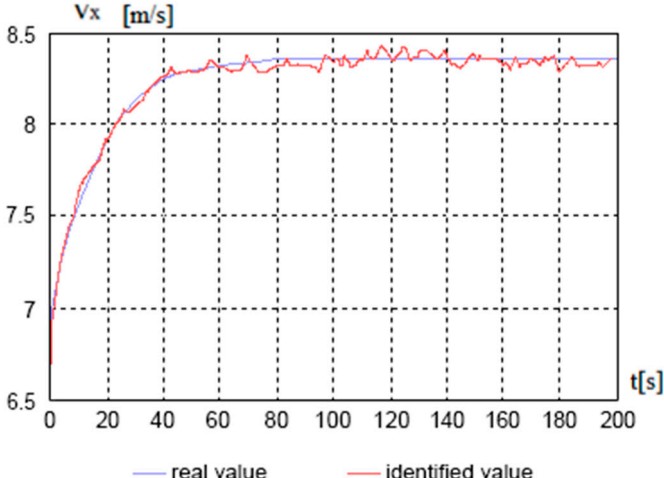

**Figure 1.** Estimation of the speed of rectilinear movement.

In Equation (1) $m_{11} = m_c + \Delta m_{11}$, $m_{22} = m_c + \Delta m_{22}$—the mass of ship with entrained water; $J_{zp} = J_{zc} + \Delta J_{zp}$—vessel's moment of inertia with attached water inertia moment; $v = \sqrt{v_x^2 + v_y^2}$, $v_1 = \sqrt{v^2 + (\omega_Z L_r)^2}$, $v_2 = \sqrt{v^2 + (\omega_Z L_r)^2}$—water flow rate modules, respectively, in the area of the ship's center of mass, bow and rudder;

$\beta = -\arcsin\frac{v_y}{v}$—leeway angle; $\rho$—water density; $q = \frac{\rho v^2}{2}$, $q_1 = \frac{\rho v_1^2}{2}$, $q_2 = \frac{\rho v_2^2}{2}$—dynamical pressure; $\overline{\omega}_Z = \frac{\omega_Z L}{v_1}$, $\overline{\omega}_r = \frac{\omega_Z L_r}{v_2}$—relative angular velocities.

Vessel parameters such as ship length $L$, width $B$, rudder from the center of mass standoff distance $L_r$, and rudder area $S_r$ can be considered to be constant. In general, hydrodynamic coefficients of forces and moments depend on the drift angle and angular velocity of the ship's rotation. To determine these for a particular vessel we can use expressions given in [4–8].

A number of ship parameters, such as $\Delta m_{11}$, $\Delta m_{22}$—entrained water, $\Delta J_{zp}$—added mass moment of inertia of water, and $T_{oc}$—ship's draft are random variables. These random parameters can be defined as some calculated (nominal) non-random values and random deviations from the calculated values. The controllability characteristics of each specific ship will clearly depend on the realized random parameters of the ship.

Let us accept that model parameters $T_x$, $K_v$, $C_{x0}$ and $m_y$ are random variables with equally probable distribution laws in the intervals $[T_{x\min}, T_{x\max}]$, $[K_{v\min}, K_{v\max}]$, $[C_{x0\min}, C_{x0\max}]$, $[m_{y\min}, m_{y\max}]$. All other parameters are known values.

To estimate the random parameters, $T_x$, $K_v$, $C_{x0}$ of the model we perform a vessel test in rectilinear motion ($\delta(t) = 0$) at full ($n_v = n_{\max}$), half ($n_v = n_{\max}/2$), and slow ($n_v = n_{\max}/4$) speeds. During the on-board tests, the ship's speed $v_x^*(t_i)$, acceleration $a_x^*(t_i)$, and measurement times $t_i$ are measured and recorded. The equations of rectilinear motion model (considering $v_y(t) = 0$, $\beta(t) = 0$, $\omega(t) = 0$) can be written in the form:

$$\frac{dv_x}{dt} = a_x(t) = \left(\frac{n_v^2}{m_{11}}\right)T_x + \left(\frac{n_v}{m_{11}}\right)v_x K_v + \frac{C_{x0}\rho S_{cp} v_x^2}{2m_{11}} \tag{2}$$

The measurement model can be represented as:

$$v_x^*(t_i) = v_x(t_i) + \sigma_v \xi_{vx}(t_i), \ a_x^*(t_i) = a_x(t_i) + \sigma_a \xi_{ax}(t_i) \tag{3}$$

where $\sigma_v$, $\sigma_a$ are the measurement's MSE; $\xi_{vx}(t_i)$, $\xi_{ax}(t_i)$ are independent white noises.

To estimate the random parameter $m_y$ of the model (with known estimates $\hat{T}_x$, $\hat{K}_v$, $\hat{C}_{x0}$) we test the vessel during circulation ($\delta(t) = \delta_m$, at full speed $n_v = n_{\max}$). The model motion during circulation equations, according to Equation (1), are written in the form:

$$\begin{aligned}
\frac{dv_x}{dt} &= a_x(t) = F_x\big(\omega, \delta_m, v_x, v_y, \hat{T}_x, \hat{K}_v, \hat{C}_{x0}\big), \\
\frac{dv_y}{dt} &= a_y(t) = F_y\big(\omega, v_x, v_y\big) + m_y f\big(\omega, \delta_m, v_x, v_y\big), \\
\frac{d\omega}{dt} &= a_\omega(t) = M\big(\omega, v_x, v_y\big) + m_y f\big(\omega, \delta_m, v_x, v_y\big) L_r m_{22}/J_Z.
\end{aligned} \tag{4}$$

During circulation, in addition to the measurements of Equation (3), lateral velocity and acceleration, angular velocity, and angular acceleration of the ship's rotation are also measured [2,3,7–9]:

$$\begin{aligned}
v_y^*(t_i) &= v_y(t_i) + \sigma_v \xi_{vy}(t_i), \ a_y^*(t_i) = a_y(t_i) + \sigma_a \xi_{ay}(t_i), \\
\omega^*(t_i) &= \omega(t_i) + \sigma_\omega \xi_\omega(t_i), \ a_\omega^*(t_i) = a_\omega(t_i) + \sigma_\omega \xi_{a\omega}(t_i)
\end{aligned} \tag{5}$$

Based on the measured natural information of Equation (3) and, accordingly, Equation (5), it is possible, using simple linear estimation algorithms [5], to construct estimates of velocities $\hat{v}_x(t_i)$, $\hat{v}_y(t_i)$ and angular velocity $\hat{\omega}(t_i)$. A linear estimation algorithm, for example, for velocity $v_x(t)$ (similarly $v_y(t)$, $\omega(t)$) is determined by the equations:

$$\begin{aligned}
K_v(t_i|t_{i-1}) &= K_v(t_{i-1}) + \sigma_a^2, \ P_v(t_i) = K_v(t_i|t_{i-1})/\big(K_v(t_i|t_{i-1}) + \sigma_v^2\big), \\
K_v(t_0) &= \sigma_v^2
\end{aligned}$$

$$\begin{aligned}
\hat{v}_x(t_i) &= \hat{v}_x(t_{i-1}) + a_x^*(t_{i-1})dt + P_v(t_i)(v_x^*(t_i) - \hat{v}_x(t_i)), \\
K_v(t_i) &= K_v(t_i|t_{i-1}) - P_v(t_i)K_v(t_i|t_{i-1})
\end{aligned} \tag{6}$$

where $K_v(t_i)$ is the velocity estimation variance and $dt = t_i - t_{i-1}$ is the measurement arrival time interval. Realizations of the estimates of speeds and angular velocity when simulating the rectilinear motion of a Volgo-Balt-type vessel are shown in Figures 1–4 at full speed (Figure 1) and during circulation (Figures 2–4). Measurement accuracies are $\sigma_v = 0.1$ m/s, $\sigma_a = 0.01$ m/s$^2$, time interval $dt = 0.5$ s.

Based on estimates of the speed and angular velocity of the vessel's circulation, it is possible to calculate the functions in Equations (2) and (4).

$$\begin{aligned}
f_K(\hat{v}_x(t)) &= \left(\frac{n_v}{m_{11}}\right)\hat{v}_x(t), \ f_{Cx0}(\hat{v}_x(t)) = \frac{\rho S_{CP}\hat{v}_x^2(t)}{2m_{11}}, \ F_y\big(\hat{\omega}(t), \hat{v}_x(t), v_y(t)\big), \\
&M\big(\hat{\omega}(t), \hat{v}_x(t), \hat{v}_y(t)\big), \ f\big(\hat{\omega}(t), \delta_m, \hat{v}_x(t), \hat{v}_y(t)\big).
\end{aligned} \tag{7}$$

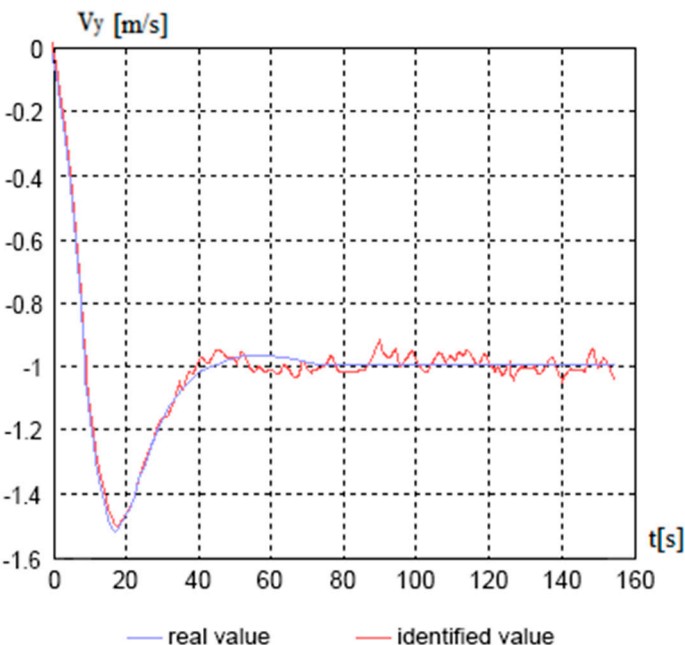

**Figure 2.** Estimation of the V$_x$ component of velocity during circulation.

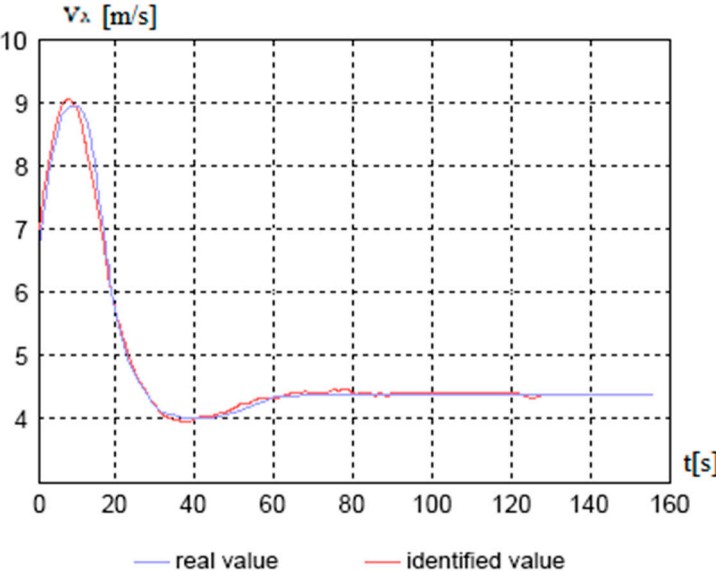

**Figure 3.** Estimation of the V$_y$ component of the velocity during circulation.

The measured acceleration $a_x^*(t_i)$, according to Equations (2) and (3) is a linear trend of random parameters $T_x$, $K_v$, $C_{x0}$:

$$a_x^*(t_i) = \left(\frac{n_v^2}{m_{11}}\right)T_x + f_K(\hat{v}_x(t))K_v + f_{Cx0}(\hat{v}_x(t))C_{x0} + \sigma_a\xi_{ax}(t_i) \tag{8}$$

The measured accelerations $a_y^*(t_i)$, $a_\omega^*(t_i)$, according to Equations (4) and (5), are linear trends of random parameter $m_y$:

$$a_y^*(t_i) = F_y\big(\hat{\omega}(t), \hat{v}_x(t), v_y(t)\big) + f\big(\hat{\omega}(t), \delta_m, \hat{v}_x(t), \hat{v}_y(t)\big)m_y + \sigma_a\xi_{ay}(t_i),$$
$$a_\omega^*(t_i) = M\big(\hat{\omega}(t), \hat{v}_x(t), \hat{v}_y(t)\big) + \frac{f\big(\hat{\omega}(t), \delta_m, \hat{v}_x(t), \hat{v}_y(t)\big)L_r m_{22}}{J_Z m_y} + \sigma_\omega\xi_{a\omega}(t_i) \tag{9}$$

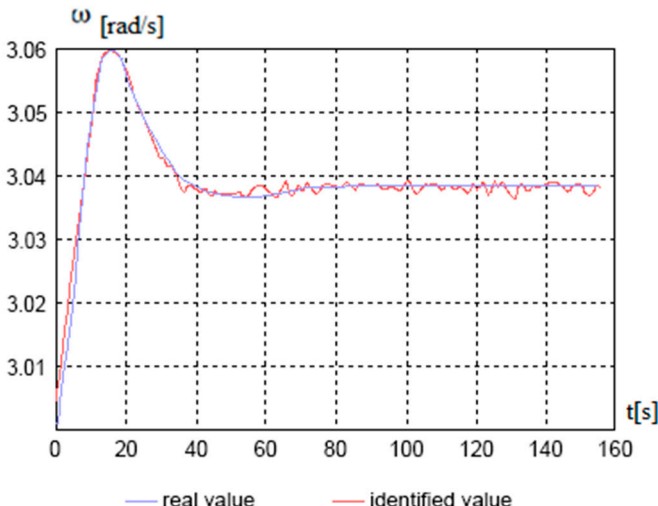

**Figure 4.** Estimation of the angular velocity of the vessel during circulation.

## 3. Synthesis of Algorithms for Estimating Ship Parameters

The estimation algorithm for a vector $X$ of random parameters by the criterion of the minimum mean square error, in the general case, is reduced to calculating the conditional mean.

$$\widetilde{X}(t) = \int_{\Omega(X)} XP(X|Y^*(\tau))\,dX \tag{10}$$

where $P(X|Y^*(\tau))$ is the after-the-event probability of the parameter vector $X$ given the measured sample of the vector $Y^*$ on the time interval $t_0 \leq \tau \leq t$; $\Omega(X)$ is the region of the possible values of vector $X$.

If there is a sufficient statistic $\hat{X}(t) = \hat{X}(Y^*(\tau))$, then the posterior probability, using Bayes' formula, can be written in the form:

$$P(X|Y^*(\tau)) = \frac{P(\hat{X}(t)|X)P(X)}{\int_{\Omega(X)} P(\hat{X}(t)|X)\,P(X)dX} \tag{11}$$

## 4. Estimation of Parameters for Rectilinear Motion

When testing the ship in straight-line motion, the measurement vector is recorded:

$$Y_p^*(t_i) = C_p(t_i)X_p + T_p(t_i)\xi(t_i) \tag{12}$$

where $C_p(t_i)$—measurement matrix; $T_p(t_i)$—RMSD matrix of measurement errors; $X_p$—vector of estimated parameters.

$$C_p(t_i) = \begin{bmatrix} n_m^2/m_{11}, \ f_K(\hat{v}_{xp}(t)), \ f_{Cx0}(\hat{v}_{xp}(t)) \\ n_m^2/4m_{11}, \ f_K(\hat{v}_{xc}(t)), \ f_{Cx0}(\hat{v}_{xc}(t)) \\ n_m^2/16m_{11}, \ f_K(\hat{v}_{xm}(t)), \ f_{Cx0}(\hat{v}_{xm}(t)) \end{bmatrix}, T_p(t_i) = \begin{bmatrix} \sigma_a & 0 & 0 \\ 0 & \sigma_a & 0 \\ 0 & 0 & \sigma_a \end{bmatrix}, X_p = [T_x, K_v, C_{x0}] \tag{13}$$

The sufficient statistic $\hat{X}_p(t)$ has a linear estimate determined by the equations:

$$K_p(t_i) = K_p(t_{i-1}) - K_p(t_{i-1})C_p(t_i)^T \begin{pmatrix} C_p(t_i)K_p(t_{i-1})C_p(t_i)^T \\ +T_p(t_i)T_p^T(t_i) \end{pmatrix}^{-1} C_p(t_i)K_p(t_{i-1}),$$

$$P_p(t_i) = K_p(t_{i-1})C_p(t_i)^T \begin{pmatrix} C_p(t_i)K_p(t_{i-1})C_p(t_i)^T \\ +T_p(t_i)T_p^T(t_i) \end{pmatrix}^{-1}, \tag{14}$$

$$\hat{X}_p(t_i) = \hat{X}_p(t_{i-1}) + P_p(t_i)\left(Y_p^*(t_i) - C_p(t_i)\hat{X}_p(t_{i-1})\right).$$

The conditional probability of a linear estimate for a sufficiently large sample of measurements can be considered a Gaussian distribution; in this case the optimal parameter estimate, according to Equation (11), will be determined by the expression [2,10,11]:

$$\widetilde{T}_x(t) = \hat{T}_x(t) + \sigma_T(t)\sqrt{\frac{2}{\pi}} \frac{\left(e^{-x_2^2} - e^{-x_1^2}\right)}{\text{erf}(x_2, x_1)} \tag{15}$$

where $x_1 = \frac{\hat{T}_x(t) - T_{\min}}{\sqrt{2}\sigma_T(t)}$, $x_2 = \frac{\hat{T}_x(t) - T_{\max}}{\sqrt{2}\sigma_T(t)}$, $\sigma_T(t) = \sqrt{K_{p11}(t)}$ $\text{erf}(x_2, x_1) = \frac{2}{\sqrt{\pi}}\int_{x_1}^{x_2} e^{-x^2}dx$ is the error function integral. The optimal estimates of the parameters $K_v$, $C_{x0}$ are determined similarly.

## 5. Estimation of Parameters during Circulation

During circulation, a measurement vector is recorded:

$$Y_{ts}^*(t_i) = F_{ts}(t_i) + C_{ts}(t_i)m_y + T_{ts}(t_i)\xi(t_i)$$

where

$$F_{ts}(t_i) = \begin{bmatrix} F_y\big(\hat{\omega}(t), \hat{v}_x(t), v_y(t)\big) \\ M\big(\hat{\omega}(t), \hat{v}_x(t), v_y(t)\big) \end{bmatrix}, C_{ts}(t_i) = \begin{bmatrix} f\big(\hat{\omega}(t), \delta_m, \hat{v}_x(t), \hat{v}_y(t)\big) \\ f\big(\hat{\omega}(t), \delta_m, \hat{v}_x(t), \hat{v}_y(t)\big)m_{22}/J_Z \end{bmatrix},$$
$$T_{ts}(t_i) = \begin{bmatrix} \sigma_a & 0 \\ 0 & \sigma_\omega \end{bmatrix} \tag{16}$$

A linear estimate of the parameter $m_y$ is determined by the equation:

$$\hat{m}_y(t_i) = \hat{m}_y(t_{i-1}) + P_{ts}(t_i)\big(Y_{ts}^*(t_i) - F_{ts}(t_i) + C_{ts}(t_i)\hat{m}_y(t_{i-1})\big) \tag{17}$$

where the vector of weights $P_{ts}(t_i)$ is defined similarly to Equation (14).

The optimal estimate of the parameter $m_y$ is determined by the expressions:

$$\widetilde{m}_y(t) = \hat{m}_y(t) + \sigma_m(t)\sqrt{\frac{2}{\pi}} \frac{\left(e^{-x_2^2} - e^{-x_1^2}\right)}{\text{erf}(x_2, x_1)}$$

where

$$x_1 = \frac{\hat{m}_y(t) - m_{y\min}}{\sqrt{2}\sigma_m(t)}, x_2 = \frac{\hat{m}_y(t) - m_{y\max}}{\sqrt{2}\sigma_m(t)}, \sigma_m(t) = \sqrt{K_{ts}(t)}. \tag{18}$$

## 6. Accuracy Analysis of the Estimation Algorithms

Accuracy analysis of the synthesized algorithms for estimating the ship parameters was carried out using a statistical modeling method. A Volgo-Balt type vessel was considered with the main known movement model parameters of Equation (1), as shown in Table 1.

**Table 1.** Movement model parameters of Equation (1) of the Volgo-Balt type vessel.

| Name | $m_{11}[\text{tf}*\text{s}^2/\text{m}]$ $m_{22}[\text{tf}*\text{s}^2/\text{m}]$ | $J_z[\text{tf}*\text{s}^{2*}\text{m}]$ $l_{26}[\text{tf}*\text{s}^2]$ | L [m] $L_r$ [m] | $C_r$ [m²] $S_r$ [m²] | $A_p$ [m²] $S_{cp}$ [m²] | $m_1$ $m_2$ | $C_\beta$ $C_{\omega\beta}$ | $C_{\omega\omega}$ $n_v$[r/s] |
|------|------|------|------|------|------|------|------|------|
| value | 232 380.26 | 133,630 −539.91 | 81.7 40.5 | 0.942 8.1 | 331.7 1065 | 0.0604 0.003378 | 0.283 0.1576 | −0.097836 4.8 |

The Volgo-Balt vessel is a dry-cargo vessel of the "river-sea" class, designed for the carriage of bulk cargo (coal, ore, grain, crushed stone, etc.) along a country's large inland waterways with access to the sea.

The implementation of estimation of the vessel's parameters with time is shown in Figures 5–8 (the number of measurements $n_3 = 2t$).

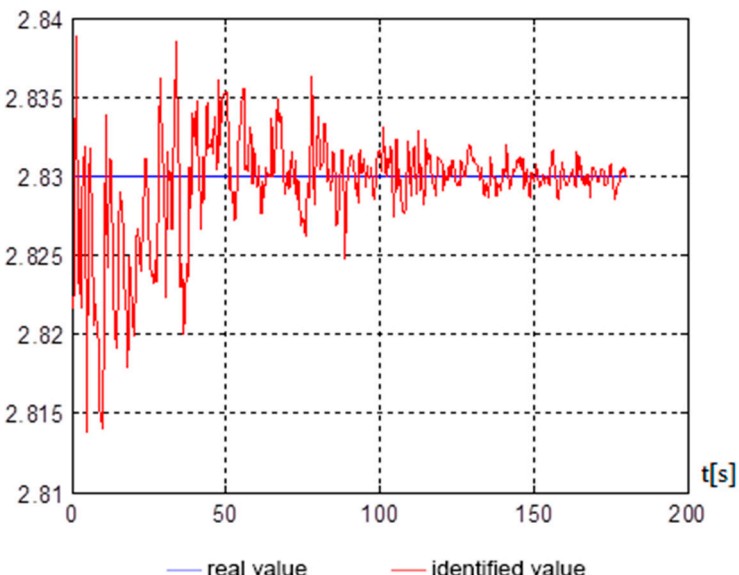

**Figure 5.** $T_x$ assessment over time.

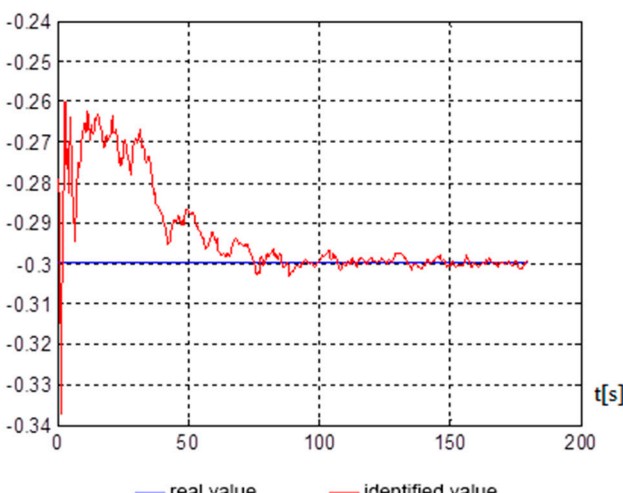

**Figure 6.** $K_v$ assessment over time.

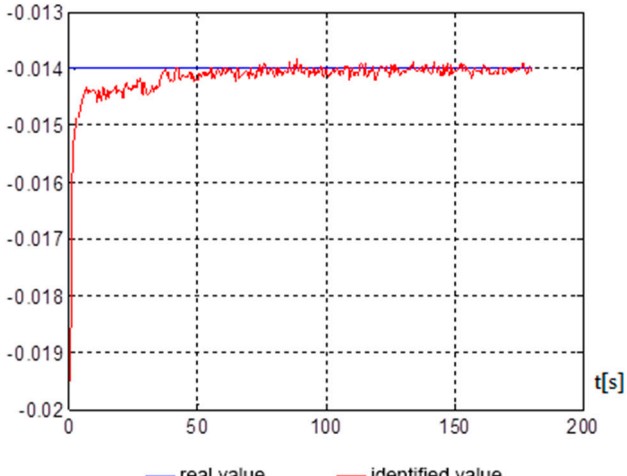

**Figure 7.** $C_{x0}$ assessment over time.

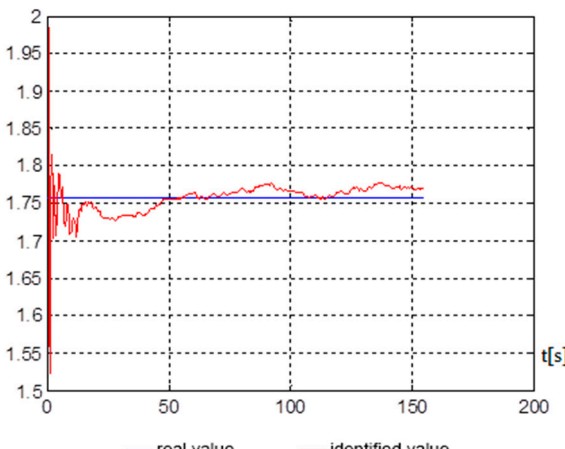

**Figure 8.** $m_y$ assessment over time.

Figures 5–8 show, on average, at 150 s, that good indicators are obtained for evaluating the parameters, namely: $\sigma_T = 0,0764\,\text{tf}\cdot\text{s}^2/\text{rev}^2$, $\sigma_K = 0,058\,\text{tf}\cdot\text{s}^2/\text{rev}^2\cdot\text{m}$, $\sigma_C = 0.0005$, $\sigma_m = 0.046$. The end of the identification transition process occurs at 50 s. For a vessel that is an inertial object, the resulting identification speed is more than satisfactory and can be used in automated control systems of a marine vessel.

For identification, parameters were selected that significantly affect the dynamics of the ship's movement [11]. Furthermore, the obtained algorithms can be used for all other parameters that cannot be accurately determined from analytical calculations.

## 7. Conclusions

This paper considers the problem of synthesizing algorithms for estimating the ship motion model parameters, based on measured information in field tests. The derived algorithms are relatively simple and allow highly precise estimates of the unknown parameters of a ship's motion model at a non-real-time scale to be obtained using measurements recorded in field tests. The results can be used in the construction of automated ship control systems, and in the development of navigation simulators, for the creation of ship models.

The algorithms developed in this work were tested when navigating a vessel of the "Volgo-Balt" type project 2-95A/R through the Kerch Strait. The mathematical model obtained after identifying the parameters showed good convergence with a full-scale prototype. On the basis of the obtained model, an assessment was made of the risk of escorting the vessel at each section of the strait, and an algorithm for changing the rudder was recommended, taking into account hydrometeorological factors.

**Author Contributions:** Conceptualization, A.Z. and S.G.C.; methodology, N.I.; software, V.E.; validation, N.I., S.G.C. and A.Z.; formal analysis, N.I.; investigation, A.Z.; resources, N.I.; data curation, S.G.C.; visualization, V.E.; supervision, N.I.; project administration, S.G.C. All authors have read and agreed to the published version of the manuscript.

**Funding:** The research is partially funded by the Ministry of Science and Higher Education of the Russian Federation as part of the World-class Research Center program: Advanced Digital Technologies (contract No. 075-15-2020-903 dated 16 November 2020).

**Institutional Review Board Statement:** Not applicable.

**Informed Consent Statement:** Not applicable.

**Data Availability Statement:** Not applicable.

**Conflicts of Interest:** The authors declare no conflict of interest.

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
