# Peer review of "Development of Algorithms for Identifying Parameters of the Maritime Vessel Motion Model in Operating Conditions with Elements of Intellectual Analysis"

_jmse, doi:10.3390/jmse9040418_

Round 1
Reviewer 1 Report
Authors propose a technique for gathering estimates of certain parameters of ship motion which are required for building computerized ship models for simulators or ship control construction. The methods are basically linear estimates with noise, based on observed values. The problem is relevant, but it is not clear to me how innovative this technique is. I am not an expert in this particular area; linear estimates of this sort would seem to be an obvious strategy.
I made numerous comments and emendations suggested in the attached pdf. Many concerned English.
In the many graphs there was no legend describing the different color lines. Graphs need to be fully explained.
Authors should insure the entire paper is in a single font and size, including the references.
Page 1 line 21ff. The last line of the Abstract is not a sentence and is confusing. Perhaps just leave it out.
Page 3 line 84 Authors refer to flat noise. I am unfamiliar with that term. Do they mean white noise?
Page 10 figure 8. Why is there a drift upward at the end? Is this to be expected, or would measured values come to the blue line in most cases?
Page 10 line 231. Parameter error estimates are given, but you need to add a paragraph discussing whether these are reasonable, and how they compare with other techniques. Otherwise they are only partially useful, and the heart of the paper is (or should be) that it is better than some previously employed method, or guesswork.
I wonder if there are some US or European papers on this subject.

Author Response
Dear Reviewer.
We are very grateful to you for such a serious work with our article. Your comments significantly improved the content of the article and made it better. We changed all the graphics and added a legend.
Formulas have been processed and mathematical inaccuracies have been removed.
The list of references has been changed.
References to the literature have been added throughout the text of the article.
All adjustments have been highlighted. We took into account all your comments. Thank you for such a serious and good job.
We have attached all the changes to the article file.
Reviewer 2 Report
- Well interested items.
- However, the article was hard to read and was not nice to understand.
- Also, fonts for every paragraph were differ to each other. So authors should recheck and unify every fonts.
- Authors should explain basic equations of motions and matrix for components in the equations (i.e. equations/matrix for m11~m66)
- Authors should show coordinates which they have used.
- Explanation for components in the equations, already described in the article (i.e. page 3, line 86) was hart to read / unkind for readers.
- Figures 1-4: authors have to show legends for each data and should leave comments for each figures.
Therefore, I would suggest "Reconsider after major revision".
Author Response

(The authors gave the same response as above.)

Round 2
Reviewer 1 Report
Just a couple of small changes. Make sure the formulas fit on the pages. See the attached markup pdf.

Author Response
Thank you for your comments. This made it possible to do the better work. We have made a correction to the formulas. Allowed for formatting. Adding new information and making corrections that are highlighted in color.
Thank you for your work to improve the article.

Reviewer 2 Report
Appropriate measures were not taken for the points pointed out in the first review.
- The size of the text in the article is still different.
- The explanations for each term that follows after each formula are presented is still written in a form that is difficult to read.
- Prior to entering the detailed equations of motion, no improvements were made to the description of the basic coordinate system, and co.
- The conclusion is too short to summarize and to present the result of this study.
Author Response
Thank you for your comments. This made it possible to do a better work. We have made a correction to the formulas. Allowed for formatting. Adding new information and making corrections that are highlighted in color.
Mathematical models have been used by scientists: Voytkunsky et al., 1973; Hoffman, 1988; Pavlenko, 1979; Sobolev, 1976; Tumashik, 1978; Fedyaevsky, Sobolev, 1963.
Therefore, the reviewer can be sure that all errors have been corrected.
Experiments were made in the Matlab modeling program. This program is very well known in the modeling environment and enjoys great authority.
Thank you for your work to improve our article.

Round 3
Reviewer 2 Report
Well improved than before. But still, the authors need to correct/add something:
- page1 line 21: There are so many 'double blanked' sentences in the entire paper. I think this kind of mistake shouldn't be repeated.
- page2 line 93: Are you sure that this theory is famous in the world? The expression 'The world-famous...' is improper, I think.
- page3 line 98 (equation 1): I have pointed out constantly to add descriptions of the coordinate systems. Do you think readers will be able to know which direction the x-axis is and which direction the y-axis is?
- page9 line 237: Line/paragraph alignment is inconstant.
- page10 line 282: Then you have to explain what is 'Volga-Balt type'.
Author Response
Dear Reviewer
Thank you for your comments. We made corrections.
- We did correction " page1 line 21: There are so many 'double blanked' sentences in the entire paper. I think this kind of mistake shouldn't be repeated."
- Based on the materials LLOYD'S REGISTER (https://www.lr.org/en/) and IMO (https://www.imo.org/), conclusions can be drawn about the relevance of mathematical models."page2 line 93: Are you sure that this theory is famous in the world? The expression 'The world-famous...' is improper, I think."
- The standard notation for the x and y axes will be used for 2D space. For figure 1 defined as x = Vx [m/s], y = t [s]. " page3 line 98 (equation 1): I have pointed out constantly to add descriptions of the coordinate systems. Do you think readers will be able to know which direction the x-axis is and which direction the y-axis is? "
- We did "page9 line 237: Line/paragraph alignment is inconstant. "
- Volgo-Balt - dry-cargo vessels of the "river-sea" class, designed for the carriage of bulk cargo (coal, ore, grain, crushed stone, etc.) along the country's large inland waterways with access to the sea. The necessary explanations were made in the article"page10 line 282: Then you have to explain what is 'Volga-Balt type'."

Round 4
Reviewer 2 Report
It has been appropriately corrected for the requested matters. Please check the case, typos, grammar, etc again prior to submit the final version.This manuscript is a resubmission of an earlier submission. The following is a list of the peer review reports and author responses from that submission.